# Viral load non-suppression status among women exposed to Dolutegravir-based versus Efavirenz-based regimens in Ethiopia: A before-and-after study

Wolde Facha[1][☯]*, Takele Tadesse[1][☯], Eskinder Wolka[1][☯], Ayalew Astatkie[2][☯]

1 Department of Epidemiology and Biostatistics, School of Public Health, College of Health Sciences and Medicine, Wolaita Sodo University, Wolaita Sodo, Ethiopia, 2 School of Public Health, College of Medicine and Health Sciences, Hawassa University, Hawassa, Ethiopia

☯ These authors contributed equally to this work.
* woldie.facha@wsu.edu.et, woldiefacha@gmail.com

**Data Availability Statement:** All relevant data are within the manuscript and its Supporting Information files.

## Abstract

### Background

High viral load during pregnancy and breastfeeding period is the risk factor for vertical transmission of human immunodeficiency virus (HIV). Currently, Dolutegravir (DTG)-based regimens are recommended to attain adequate viral load suppression (VLS) among women. However, its effect on VLS has not been investigated among women in PMTCT care in Ethiopia.

### Objective

This study aimed to investigate the rate of viral load non-suppression among women exposed to DTG-based versus Efavirenz (EFV)-based regimens in Ethiopia.

### Methods

An uncontrolled before-and-after study design was conducted among 924 women (462 on EFV-based and 462 on DTG-based regimens) enrolled in PMTCT care from September 2015 to February 2023. The outcome variable was the viral load (VL) non-suppression among women on PMTCT care. A modified Poisson regression model was employed, and the proportion was computed to compare the rate of VL non-suppression in both groups. The risk ratio (RR) with a 95% confidence interval (CI) was calculated to assess viral load non-suppression among women on DTG-based and EFV-based regimens by adjusting for other variables.

### Results

The overall rate of non-suppressed VL was 16.2% (95% CI: 14.0–18.8%). Mothers on DTG-based regimens had approximately a 30% (adjusted risk ratio (aRR): 0.70; 95% CI: 0.52–0.94) lesser risk of developing non-suppressed VL than women on EFV-based regimens.

**Funding:** The author(s) received no specific funding for this work.

**Competing interests:** The authors have declared that no competing interests exist.

**Abbreviations:** ART, Antiretroviral Therapy; ARV, Antiretroviral; aRR, Adjusted Risk Ratio; CI, Confidence Interval; cRR, Crude Risk Ratio; DTG, Dolutegravir; EFV, Efavirenz; HIV, Human Immunodeficiency Virus; IQR, Interquartile Range; MTCT, Mother to Child Transmission; PMTCT, Prevention of Mother to Child Transmission; VL, : Viral Load; VLS, Viral Load Suppression; WHO, World Health Organization.

Besides, older women were 1.38 times (aRR: 1.38; 95% CI: 1.04–1.83); mothers who did not disclose their HIV status to their partners were 2.54 times (aRR: 2.54; 95% CI: 1.91–3.38); and mothers who had poor or fair adherence to antiretroviral (ARV) drugs were 2.11 times (aRR: 2.11; 95% CI: 1.45–3.07) at higher risk of non-suppressed VL.

## Conclusion

Women on DTG-based regimens had a significantly suppressed VL compared to those on EFV-based regimens. Thus, administering DTG-based first-line ART regimens should be strengthened to achieve global and national targets on VLS.

## Introduction

Human immunodeficiency virus (HIV) remains a leading cause of morbidity and mortality among women of reproductive age and their children [1]. Globally, approximately 39 million people are living with HIV, and 53% of them were women and girls in 2022 [2]. In this year, 82% of pregnant women living with HIV had access to antiretroviral therapy (ART) to prevent mother-to-child transmission (MTCT) of HIV [3]. Globally, approximately 130,000 children (0–14 years old) acquired the virus by the end of 2022 [4]. The number of new HIV infections among children (0–14 years) was approximately 2,000 in Ethiopia and 172 in study regions in 2022 [5, 6]. Most infections occur during pregnancy, labour and delivery, and breastfeeding periods due to a non-suppressed maternal viral load [7, 8].

Identifying women's HIV status at early pregnancy, putting women on ART, preferably DTG-based regimens, and retaining them in care complemented with good adherence are key interventions to avert the risk of MTCT of HIV. Viral load testing is the gold standard for HIV treatment monitoring and is recommended as the preferred approach to assess the success of ART [9, 10]. A high viral load during pregnancy and breastfeeding is the major risk factor for MTCT of HIV [11, 12] and becomes a challenge to the global plan to end the AIDS epidemic by 2030 [13]. Instead of the previous EFV-based regimens, the World Health Organization (WHO) recommended Integrase Strand Transfer Inhibitors (InSTIs), the DTG-based regimens, as the preferred first-line antiretroviral (ARV) drugs for pregnant and breastfeeding women as of July 2019 [14, 15]. The recommendation for the change in regimens by WHO was a result of better tolerability, a higher genetic barrier to resistance, lower adverse events, lower cost, and rapid VLS by the DTG-based regimens compared to the previously used EFV-based regimens [12, 15–18].

As per the United Nations Program on HIV/AIDS (UNAIDS) global targets, Ethiopia has planned to achieve the third 95% target (95% of people receiving ART have achieved an undetectable VL) since 2020 [19, 20]. Except for five African countries, other developing countries including Ethiopia did not achieve the third 95% target among women [11, 21–32]. Besides, previous studies focused on VLS among women other than DTG-based regimens and the safety and efficacy of DTG-based versus EFV-based regimens among people living with HIV. However, the effectiveness of DTG-based regimens on VLS over the previously used EFV-based regimens has not been investigated among women in PMTCT care in Ethiopia. Therefore, this study aimed to investigate the rate of viral load non-suppression and its risk factors among women exposed to DTG-based versus EFV-based first-line antiretroviral therapy in Ethiopia.

## Materials and methods

### Study setting

The study was conducted in two regions of Ethiopia: Central Ethiopia and South Ethiopia (S1 Fig). The South Ethiopia region is administratively divided into 12 zones (the largest administrative structure in the region), whereas Central Ethiopia is divided into seven zones and three special districts [33, 34]. The seats of these zones and special districts are categorized into nine high-, ten medium-, and three low-priority towns based on the incidence of HIV [35]. In these regions,140 health facilities (49 hospitals and 91 health centers) are providing ART and PMTCT services to 1,236 pregnant and breastfeeding women (675 in South Ethiopia and 561 in Central Ethiopia). However, only 40 hospitals and 32 health centers were have been providing both PMTCT and ART services since 2015, when the study started. Thirty-four facilities (20 hospitals and 14 health centers) from the nine high-priority and ten medium-priority towns were selected among 72 facilities that have been offering PMTCT services.

### Study design, period, and participants

An uncontrolled before-and-after study design was conducted among 924 women (462 on EFV-based and 462 on DTG-based regimens) enrolled in the PMTCT care from September 2015 to February 2023. The source population for the unexposed (before) group was all women on EFV-based first-line ART, and for the exposed (after) group, it was all women on DTG-based first-line ART in Ethiopia. The study participants for the unexposed group were all eligible women enrolled in PMTCT care from September 2015 onwards and took only EFV-based regimens until discharge, whereas eligible women who took only DTG-based regimens for the entire PMTCT period until discharge were included in the exposed group. However, those women who were on ART for less than three months, who had no viral load test result, and who shifted the drugs from EFV-based regimens to DTG-based regimens were excluded from the study. Women in high- and medium-priority areas were considered for inclusion, and those in low-priority areas were excluded as they either have not started the PMTCT service since September 2015 (when the study started) or were few (below 12) women in PMTCT care due to resource limitations.

### Sample size determination and sampling technique

The sample size was calculated using the double population proportion formula using G Power version 3.1.9.7 statistical software. A significance level (alpha) of 5%, a power of 80%, the proportion of non-suppressed VL among women on DTG-based regimens of 20.3%, the proportion of non-suppressed VL among women on EFV-based regimens of 29% based on a study conducted in Cameroon [26], and the ratio of unexposed to exposed of one. After adding 20% due to experts' suggestions to compensate for the expected missing data, the total sample size was 924 (462 for exposed and 462 for unexposed groups). The final sample size was allocated proportionally to the number of women enrolled in PMTCT care in the study area. A total of 72 facilities providing both ART and PMTCT services since 2015 were stratified by facility type (hospitals and health centers) and priority towns (high, medium, and low) based on the incidence of HIV [35]. Then, a simple random sampling was employed in each stratum to select 20 hospitals and 14 health centers. Since facilities under each category have similar characteristics in terms of service quality, women's characteristics, and the level of staff providing the service, all eligible women in selected facilities were included in the study starting from those discharged from PMTCT care in the most recent period and went backward in time selecting study participants serially until the required sample size was met.

## Operational definitions

**Exposed group.**   Pregnant and/or breastfeeding women who were on DTG-based first-line ART until discharge from PMTCT care.

**Unexposed group.**   Pregnant and/or breastfeeding women who were on EFV-based first-line ART until discharge from PMTCT care.

**Viral load suppression.**   A viral load count of less than or equal to 50 copies/ml at any time during the PMTCT period [36, 37].

**Low-level viremia.**   One or more viral load results that are detectable (more than 50 copies/ml) but equal to or less than 1000 copies/ml [36, 37].

**Virological failure.**   A viral load above 1000 copies/ml based on two consecutive viral load measurements in 3 months, apart with enhanced adherence support following the first viral load test [36, 37].

**Viral load non-suppression.**   A viral load count of greater than 50 copies/ml at any time during the PMTCT period.

## Study variables

The outcome variable was the viral load suppression status (categorized as suppressed and non-suppressed VL) measured after three months of ART initiation among pregnant or breastfeeding women. Currently, ART efficacy among women on PMTCT care is assessed after three months of therapy [12]. The exposure variable was the ART regimen the mother was receiving. The covariates in the present study were maternal socio-demographic, obstetric, drug, and clinical-related characteristics. Maternal socio-demographic characteristics included age, residence, marital status, educational status, and occupation. Obstetric-related characteristics included antenatal care, syphilis test, and place of delivery. Drug and clinical-related characteristics included enrolment type, WHO clinical stage, adherence status, disclosure status, partner HIV status, duration of PMTCT care, timing of ART initiation, and type of health facility. Mothers' ART adherence was categorized as *poor, fair*, or *good*. Women with poor adherence status at any time during the follow-up period were classified as having *poor adherence.* It is considered poor if a woman missed greater than or equals to five out of the 30 doses, or greater than or equals to 10 out of the 60 doses at any time during the follow-up period. A woman whose adherence status was recorded as fair (but not poor at any time during the follow-up period) was classified as having *fair adherence.* This means that a woman missed 2–4 out of the 30 doses or 4–9 out of the 60 doses. On the other hand, a woman whose adherence status was recorded as good (but not poor or fair at any time during the follow-up period) was classified as having *good adherence.* It was recorded as good if a woman missed only one out of the 30 doses or less than or equals to three out of 60 the doses [12, 36].

## Statistical analysis

Data were retrieved from the PMTCT registration book, Smart Care (a computer-based data registry found at the ART unit of respective facilities), and women's folder (intake form and follow-up card). Data were collected using Open Data Kit (ODK) version 2.4 and then exported to Stata 14.0 (StataCorp, College Station, Texas, U.S.A.) for analysis. Descriptive statistics (median and interquartile range) were calculated for continuous data, and frequencies, and percentages were calculated for categorical data. The demographic, obstetric, drug and clinical characteristics across the ART treatment groups were assessed using Pearson's chi-squared test. A bivariate modified Poisson regression model was employed to select covariates for entry into the multivariable model. Covariates associated with the outcome variable (non-suppressed VL) at $p < 0.25$ in the unadjusted analyses were selected for multivariate analysis.

Multivariate risk ratios (RRs) and their 95% confidence intervals (CIs) were estimated for risk factors using the modified Poisson regression model. Besides, maternal educational status is considered for its practical importance and entered into the multivariate model [38]. Potential confounders (residence and facility type) were stratified during analysis, but there were no confounding effects since the crude result does not differ from the adjusted result.

## Ethical considerations

Ethical approval and clearance were obtained from the Institutional Review Board (IRB) of the College of Health Sciences and Medicine, Wolaita Sodo University (ethical approval number WSU41/32/223) (S1 File). Since secondary data were used, informed consent for study participants was waived by IRB ethical clearance. The confidentiality of patient-related data was maintained by avoiding possible identifiers, such as the names of the mothers; only numerical identification was used as a reference.

## Results

### Socio-demographic characteristics of the study participants

We approached a total of 1,063 women on EFV-based and DTG-based regimens and 139 (13.1%) participants' records were excluded from the final analysis due to missing data. Thus, our study included 924 women (462 in the DTG-based regimens arm and 462 in the EFV-based regimens arm) enrolled in PMTCT care at 34 selected facilities (20 hospitals and 14 health centres) in Ethiopia. In this study, 74% of women in the DTG-based regimens arm and 70.6% of women in the EFV-based regimens arm resided in urban areas. However, 35.5% of women in DTG-based regimens and 42% of women in EFV-based regimens arm did not attend formal education (Table 1).

### Obstetric characteristics

This study showed that 14.1% of women in DTG-based regimens arm and 10.0% of women in EFV-based regimens arm did not attend antenatal care during their pregnancy. However, 92.9% of women in the DTG-based regimens arm and 91.3% of women in the EFV-based regimens arm delivered their infants at a health facility (Table 2).

### Drug and clinical-related characteristics

In our study 6.3% of women in the DTG-based regimens arm and 7.8% of women in the EFV-based regimens arm had poor or fair adherence to ART during the PMTCT period. Besides, 16.0% of women in the DTG-based regimens arm and 20.4% of women in the EFV-based regimens arm did not disclose their HIV status to partners. The study also showed that 29.0% versus 28.8% of women enrolled in PMTCT care newly, and 5.6% versus 7.8% of women started ART during the delivery or breastfeeding period among women in the DTG-based and in the EFV-based regimens arm respectively. The median (IQR) maternal ART duration on PMTCT care was 22 (19–24) months for the DTG-based regimens arm and 22 (19–25) months for the EFV-based regimens arm (Table 3).

### Viral load non-suppression

The rate of viral load non-suppression among women on PMTCT care was 12.8% (95% CI: 10.0–16.2%) in the DTG-based regimens arm and 19.7% (95% CI: 16.3–23.6%) in the EFV-based regimens arm, with an overall non-suppression rate of 16.2% (95% CI: 14.0–18.8%).

**Table 1. Socio-demographic characteristics of the women in PMTCT care, Ethiopia, 2023.**

| Variables | Total (n = 924) | DTG-based regimens arm (n = 462) | EFV-based regimens arm (n = 462) | P-value[†] |
|---|---|---|---|---|
| **Age (in years)** | | | | |
| 15–29 | 513(55.5) | 257(55.6) | 256(55.4) | 0.947 |
| 30–45 | 411(44.5) | 205(44.4) | 206(44.6) | |
| **Residence** | | | | |
| Rural | 256(27.7) | 120(26.0) | 136(29.4) | 0.240 |
| Urban | 668(72.3) | 342(74.0) | 326(70.6) | |
| **Occupation** | | | | |
| High risk* | 164(17.7) | 89(19.3) | 75(16.2) | 0.228 |
| Low risk** | 760(82.3) | 373(80.7) | 387(83.8) | |
| **Educational status** | | | | |
| Formal | 566(61.3) | 298(64.5) | 268(58.0) | 0.043 |
| Not formal | 358(38.7) | 164(35.5) | 194(42.0) | |
| **Marital status** | | | | |
| Divorced/Widowed | 118(12.8) | 61(13.2) | 57(12.3) | 0.693 |
| Married | 806(87.2) | 401(86.8) | 405(87.7) | |

*High risk: commercial sex workers & daily labourers

**Low risk: housewife, employee, merchant, student

[†]Pearson chi-square test

## Factors associated with viral load non-suppression

In our study, from a total of sixteen variables, seven variables (PMTCT drug regimen, maternal age, maternal education, facility type, partner HIV status, disclosure status, and adherence status) with p <0.25 were selected as candidate variables for multivariate analysis to get a better fitting model. In multivariable analysis, women on DTG-based regimens had a 30% (aRR: 0.70; 95% CI: 0.52–0.94) lesser risk of experiencing non-suppressed VL than women on EFV-based regimens. In this study, older women were 1.38 times (aRR: 1.38; 95% CI: 1.04–1.83); women who had not disclosed their HIV status to their partners were 2.54 times (aRR: 2.54; 95% CI: 1.91–3.38); and women who had poor or fair adherence to ART drugs were 2.11 times

**Table 2. Obstetric characteristics of the women in PMTCT care, Ethiopia, 2023.**

| Variables | Total (n = 924) | DTG-based regimens arm (n = 462) | EFV-based regimens arm (n = 462) | P-value[†] |
|---|---|---|---|---|
| **Attended ANC*** | | | | |
| Yes | 813(88.0) | 397(85.9) | 416(90.0) | 0.055 |
| No | 111(12.0) | 65(14.1) | 46(10.0) | |
| **Tested for syphilis** | | | | |
| Yes | 753(81.5) | 379(82.0) | 374(80.9) | 0.672 |
| No | 171(18.5) | 83(18.0) | 88(19.1) | |
| **Place of delivery** | | | | |
| HF** | 851(92.1) | 429(92.9) | 422(91.3) | 0.393 |
| Home | 73(7.9) | 33(7.1) | 40(8.7) | |

*ANC: antenatal care

**HF: health facility

[†]Pearson chi-square test

**Table 3. Drug and clinical-related characteristics of the women in PMTCT care, Ethiopia, 2023.**

| Variables | Total (n = 924) | DTG-based regimens arm (n = 462) | EFV-based regimens arm (n = 462) | P-value[†] |
|---|---|---|---|---|
| **Adherence status** | | | | |
| Good | 859(93.0) | 433(93.7) | 426(92.2) | 0.368 |
| Poor or fair | 65(7.0) | 29(6.3) | 36(7.8) | |
| **Partner HIV status** | | | | |
| Negative | 266(28.8) | 148(32.0) | 118(25.5) | 0.029 |
| Positive/unknown | 658(71.2) | 314(68.0) | 344(74.5) | |
| **Disclosure status** | | | | |
| Yes | 756(81.8) | 388(84.0) | 368(79.6) | 0.088 |
| No | 168(18.2) | 74(16.0) | 94(20.4) | |
| **WHO stage** | | | | |
| Stage 1 | 889(96.2) | 446(96.5) | 443(95.9) | 0.605 |
| Stage > = 2 | 35(3.8) | 16(3.5) | 19(4.1) | |
| **Enrolment type** | | | | |
| Known | 657(71.1) | 328(71.0) | 329(71.2) | 0.942 |
| New | 267(28.9) | 134(29.0) | 133(28.8) | |
| **When ART started** | | | | |
| Before delivery | 862(93.3) | 436(94.4) | 426(92.2) | 0.189 |
| During delivery/BF* | 62(6.7) | 26(5.6) | 36(7.8) | |
| **Types of facility** | | | | |
| Health centre | 295(31.9) | 166(35.9) | 129(27.9) | 0.009 |
| Hospital | 629(68.1) | 296(64.1) | 333(72.1) | |
| **Maternal duration on PMTCT care (in months)** | | | | |
| Median (IQR)** | 22(19–25) | 22(19–24) | 22(19–25) | 0.456 |

*Breastfeeding

** Inter quartile range

[†]Pearson's chi-squared test

(aRR: 2.11; 95% CI: 1.45–3.07) at a higher risk of non-suppressed VL than their counterparts (Table 4).

## Discussion

This research is one of very few institution-based studies that aimed to assess the rate of viral load non-suppression among women exposed to DTG-based regimens over the EFV-based regimens in Ethiopia. In this study, the DTG-based regimens significantly suppressed viral load among women on PMTCT care than EFV-based regimens. Conversely, older age, non-disclosure of sero-status to partners, and poor adherence to ART were risk factors for virologic non-suppression among women on PMTCT care in Ethiopia.

This study revealed that women receiving DTG-based first-line ART had a 30% lesser risk of having non-suppressed VL than those receiving EFV-based regimens. The finding is in line with the study conducted in Ethiopia, India, and Tanzania, where DTG-based regimens demonstrate superior viral suppression compared with the EFV-based regimens [39–41]. The result is also consistent with the systematic review and meta-analysis that DTG-containing regimens were superior in viral suppression to EFV-containing regimens [42–45]. This is because of better tolerability and higher genetic barrier to ARV drug resistance accorded by the DTG-based regimens than that of the EFV-based regimens [12, 15, 16].

**Table 4. Effect of DTG-based first-line ART regimen and other covariates on VL non-suppression among women on PMTCT care, Ethiopia, 2023.**

| Variables | Viral Load Suppression | | cRR*(95% CI) | aRR†(95% CI) |
|---|---|---|---|---|
| | Non-suppressed (n = 150) | Suppressed(n = 774) | | |
| PMTCT drug regimen | | | | |
| Dolutegravir-based | 59(12.8) | 403(87.2) | 0.65(0.48–0.88)** | 0.70(0.52–0.94)† † |
| Efavirenz-based | 91(19.7) | 371(80.3) | 1 | 1 |
| Maternal age (in years) | | | | |
| 30–45 | 77(18.7) | 334(81.3) | 1.32(0.98–1.76) | 1.38(1.04–1.83) †† |
| 15–29 | 73(14.2) | 440(85.8) | 1 | 1 |
| Educational status | | | | |
| Not formal | 66(18.4) | 292(81.6) | 1.24(0.93–1.67) | 1.11(0.84–1.48) |
| Formal | 84(14.8) | 482(85.2) | 1 | 1 |
| Types of health facility | | | | |
| Health center | 54(18.3) | 241(81.7) | 1.20(0.89–1.62) | 1.25(0.93–1.68) |
| Hospital | 96(15.3) | 533(84.7) | 1 | 1 |
| Partner HIV status | | | | |
| Positive/unkown | 113(17.2) | 545(82.8) | 1.23(0.88–1.74) | 1.03(0.74–1.44) |
| Negative | 37(13.9) | 229(86.1) | 1 | 1 |
| Disclosed HIV status to partner | | | | |
| No | 56(33.3) | 112(66.7) | 2.68(2.01–3.57)** | 2.54(1.91–3.38) †† |
| Yes | 94(12.4) | 662(87.6) | 1 | 1 |
| Adherence to ART | | | | |
| Poor or fair | 21(32.3) | 44(67.7) | 2.15(1.46–3.17)** | 2.11(1.45–3.07)† † |
| Good | 129(15.0) | 730(85.0) | 1 | 1 |

*cRR: crude risk ratio

†aRR: adjusted risk ratio

**P-value < 0.05 in crude risk ratio

††P-value < 0.05 in adjusted risk ratio

In the present study, the overall rate of non-suppressed VL was 16.2%. This finding is lower than the studies conducted in West Shewa, Malawi, Kinshasa, Cameroon, Uganda, Rwanda, Zimbabwe, and South Africa [11, 23–26, 29–32]. These differences could be due to the differences in the drug regimens. Our study included study participants who were on DTG-based as well as on EFV-based regimens which have better tolerability to enhance ART adherence and subsequent VLS than only the EFV-based regimens. However, the finding is higher than the studies conducted in eastern Ethiopia, South Africa, and Uganda [22, 27, 28, 46]. The difference might be due to differences in study design, study subject, and operational definition for elevated viral load. The previous studies conducted in Uganda used either study subjects only on DTG-based regimens or a randomized control trial that monitored the effect of DTG-based regimens, while other studies conducted in South Africa and eastern Ethiopia considered a VL count of >1000 copies/ml as non-suppressed, while our study used an observational study design among women on PMTCT care and considered a VL count of >50 copies/ml for non-suppressed VL. Thus, extra effort has been considered in the study area to achieve the intended VLS among women on PMTCT care to prevent new HIV infections among infants born to HIV-infected women.

This study also showed that maternal age at enrolment to PMTCT care was a determinant factor for viral load non-suppression such that older women were approximately 1.38 times at higher risk of having non-suppressed VL than younger women. This finding is consistent with

the study conducted in Ethiopia [47], where older individuals are more likely to have treatment failure than younger ones. This might be because as age increases, immunity decreases because of reduced CD4+ T-cell counts related to aging, which enhances virologic non-suppression among older women [48]. Thus, due attention should be given to older women to suppress their viral load during PMTCT care. However, this finding is inconsistent with the studies conducted in Zimbabwe, Malawi, and Kinshasa, where individuals of younger age had a higher risk of virologic non-suppression than those of older age [11, 25, 31].

According to the current study, mothers who poorly adhered to ART were approximately two times at higher risk of developing non-suppressed VL than those with good adherence. This finding is consistent with the studies conducted in Ethiopia, Malawi, Uganda, and sub-Saharan Africa, which found that poor adherence is associated with non-suppressed VL [11, 23, 24, 46, 49]. Ensuring good adherence to ART among HIV-positive pregnant and/or breastfeeding women on PMTCT care is essential for improved immunological outcomes [12, 36]. Therefore, healthcare workers in the PMTCT unit should intensify adherence interventions to achieve the global and national VLS target of above 95%.

The present study revealed that disclosure of HIV status was an independent predictor for viral load non-suppression, such that women who did not disclose their HIV status to their partner were about 2.54 times more likely to experience non-suppressed VL than those who disclosed it. This finding is consistent with the studies conducted in Kinshasa, Rwanda, Uganda, and Zimbabwe, which found that those women who did not disclose their HIV status had non-suppressed VL compared to those who did not disclose it [25, 29–31, 50]. This might be because women who did not disclose their HIV status to their partners could not get financial support to collect ARV drugs from health facilities. Besides, they might not swallow the drugs on time due to fear of being seen by their partners, which might result in poor drug adherence and non-suppressed VL as a consequence. Thus, healthcare workers in the PMTCT unit need to emphasize on disclosure during counselling sessions to reduce viral load non-suppression among women on PMTCT care.

As per the investigators' knowledge, this is one of few studies that compared the effect of DTG-based regimens over EFV-based regimens on viral load non-suppression among women on PMTCT care. However, certain methodological limitations should be considered when using these results. First, there may be measurement and recording errors due to the nature of secondary data. Second, the VL test was not measured as per the recommended frequency among all women throughout their PMTCT care. Inclusion of data with recording errors and exclusion of participants with missing values from analysis might have underestimated or overestimated the rate of viral load non-suppression. Third, training might be a potential confounder since service delivery by trained and non-trained providers might affect the outcome status due to non-adherence to ART. However, due to the retrospective nature of the data, it was difficult to determine whether the PMTCT service was delivered all the time by trained providers or not.

## Conclusion

The study found that women on DTG-based ART regimens experienced superior virologic suppression compared to those on EFV-based regimens. Risk factors for viral load non-suppression included older maternal age at enrolment, non-disclosure of HIV serostatus to partner, and poor adherence to medication. Strengthening adherence counselling and disclosure of HIV status to partner is crucial for achieving VLS targets. The rate of viral load non-suppression might have been overestimated or underestimated due to recording errors and missing values.

## Supporting information

**S1 Fig. Map of the study area.**
(DOCX)

**S1 File. Ethical clearance.**
(PDF)

**S2 File. Human participants research checklist.**
(DOCX)

**S3 File. VLS data.**
(CSV)

## Acknowledgments

The authors acknowledge the staff of the South and Central Ethiopia Regional Health Bureaus for their technical and logistical support. Moreover, the authors sincerely thank all the data collectors and staff working in the PMTCT and ART units at the surveyed health facilities for their patience and cooperation during the entire data collection period.

## Author Contributions

**Conceptualization:** Wolde Facha, Takele Tadesse, Eskinder Wolka, Ayalew Astatkie.

**Data curation:** Ayalew Astatkie.

**Formal analysis:** Wolde Facha.

**Funding acquisition:** Wolde Facha.

**Investigation:** Wolde Facha.

**Methodology:** Wolde Facha, Takele Tadesse, Eskinder Wolka, Ayalew Astatkie.

**Project administration:** Wolde Facha, Takele Tadesse, Eskinder Wolka, Ayalew Astatkie.

**Resources:** Wolde Facha.

**Software:** Wolde Facha.

**Supervision:** Takele Tadesse, Eskinder Wolka, Ayalew Astatkie.

**Validation:** Takele Tadesse, Eskinder Wolka, Ayalew Astatkie.

**Visualization:** Wolde Facha.

**Writing – original draft:** Wolde Facha.

**Writing – review & editing:** Wolde Facha, Takele Tadesse, Eskinder Wolka, Ayalew Astatkie.

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
