## [Decision Letter · Decision Letter 0]

24 Apr 2024

PONE-D-24-09455Viral load non-suppression status among women exposed to Dolutegravir-based versus Efavirenz-based first-line antiretroviral therapy in Southern Ethiopia: A retrospective before-after studyPLOS ONE

Dear Dr. Facha,

Thank you for submitting your manuscript to PLOS ONE. After careful consideration, we feel that it has merit but does not fully meet PLOS ONE’s publication criteria as it currently stands. Therefore, we invite you to submit a revised version of the manuscript that addresses the points raised during the review process.

We look forward to receiving your revised manuscript.

Kind regards,

Moses Katbi, MD, MPH, Msc., FRSPH, DrPH

Academic Editor

PLOS ONE

Journal Requirements:

2.Your ethics statement should only appear in the Methods section of your manuscript. If your ethics statement is written in any section besides the Methods, please move it to the Methods section and delete it from any other section. Please ensure that your ethics statement is included in your manuscript, as the ethics statement entered into the online submission form will not be published alongside your manuscript.

3. We note that Figure 1 in your submission contain map images which may be copyrighted. All PLOS content is published under the Creative Commons Attribution License (CC BY 4.0), which means that the manuscript, images, and Supporting Information files will be freely available online, and any third party is permitted to access, download, copy, distribute, and use these materials in any way, even commercially, with proper attribution. For these reasons, we cannot publish previously copyrighted maps or satellite images created using proprietary data, such as Google software (Google Maps, Street View, and Earth). For more information, see our copyright guidelines: http://journals.plos.org/plosone/s/licenses-and-copyright.

   a. You may seek permission from the original copyright holder of Figure(s) 1to publish the content specifically under the CC BY 4.0 license. 

Please upload the completed Content Permission Form or other proof of granted permissions as an "Other" file with your submission

Reviewers' comments:

Reviewer's Responses to Questions

**Comments to the Author**

1. Is the manuscript technically sound, and do the data support the conclusions?

Reviewer #1: Yes

Reviewer #2: Yes

2. Has the statistical analysis been performed appropriately and rigorously? 

Reviewer #1: Yes

Reviewer #2: Yes

3. Have the authors made all data underlying the findings in their manuscript fully available?

Reviewer #1: Yes

Reviewer #2: Yes

4. Is the manuscript presented in an intelligible fashion and written in standard English?

Reviewer #1: Yes

Reviewer #2: Yes

5. Review Comments to the Author

Reviewer #1: Thank you for the opportunity to provide a review of this interesting article on the study of viral load non suppression status among women in a Southern Ethiopia community taking antiretroviral therapy in pregnancy and breastfeeding.

The research involved a retrospective sample size of 924 women-462 each on Dolutegravir based and Efavirenz -based antiretroviral therapy.

The result showed that women who were on Dolutegravir based antiretroviral therapy had a 30% lesser risk of developing non -suppressed viral load status than women who were on Efavirenz based therapy.

The study concluded that Dolutegravir based antiretroviral therapy as first line regime should be strengthened to achieve global and national targets on VL status ART regime in pregnant and breastfeeding women

Abstract: -Well summarised with clear description of the sub titles. Is it worth mentioning WHO recommendation in the second sentence in background here ?

Introduction- I think this was well written.

Materials and Methods: The title of the study says the region sampled was in Sothern Ethiopia but under study settings, two regions are mentioned, both Central and Southern Ethiopia.

Results-The analytical approach of the results is clear.

Discussion-

Abbreviations

Ethics approval and informed consent

Consent for publication

Data availability

Funding

Competing interest

Authors’ contributions

Acknowledgements

Authors’ information

References-satisfactory

Reviewer #2: 1. Summary of the research

The retrospective before-after study investigated the effect of DTG-based versus EFV-based regimens on viral load suppression among women in PMTCT of HIV in Southern Ethiopia which included 924 women enrolled in PMTCT care. The authors used a modified Poisson regression model to compare the rate of viral load non-suppression in both groups. The authors reported that women on DTG-based regimens had approximately 30% lesser risk of developing non-suppressed viral load status than those on EFV-based regimens. They also reported that older women, those non-disclosing HIV status to partners, and those with poor/fair adherence to ARV drugs were at a higher risk of non-suppressed viral load status. The authors concluded that women on DTG-based regimens had significantly suppressed viral load status than those who were on EFV-based regimens. They recommended strengthening DTG-based first-line ART regimens to achieve global and national targets on viral load status.

The authors claim that their finding that older women had a higher risk of non-suppression may be because as age increases, immunity decreases. This is however inconsistent with findings from a number of studies, except one research study in Ethiopia.

The manuscript is generally reasonably written, but there are some major methodological issues, including in the study design, sampling and data analysis that need review and address. The paper also has language and grammatical issues that need technical and language editing to correct and make the document flow better.

My overall recommendations are that the authors need to do a major review of the manuscript, especially the study design, sampling and data analysis of the methods, and presentation of the results. The manuscript also needs some editing in addition to rewrite of some sections to improve grammar and language for better flow.

2. Examples and evidence

2.1. Major issues

2.1.1. Overall study context: The authors provide the status of HIV and PMTCT, which seems to be the global situation, but they do specifically indicate this. They should mention that the data is global, and they should also provide the situation in Ethiopia. Furthermore, the data provided for children 0-14 years who acquired HIV seems to be for the year 2022, and the statement will need correction.

2.1.2. History of studies on effect of regimens on VLS: The following statement in the abstract and introduction, “However, its effect on VLS has not been investigated” seems to be a broad claim. It would be more accurate to say that it has not been investigated in the specific context of the study, which is among women in Southern Ethiopia or in the PMTCT care.

2.1.3. Effectiveness of DTG-based regimens: The sentence in the introduction, starting “However, the effectiveness of DTG-based regimens ….” may not be correct. Five African countries (Eswatini, Botswana, Rwanda, Tanzania and Zimbabwe) have already reached 95-95-95. (Reference: “The path that ends AIDS: UNAIDS Global AIDS Update 2023”. Geneva: UNAIDS; 2023).

2.1.4. Issue with the study design: The study design is not clearly defined. A retrospective before-after study design typically involves comparing outcomes before and after an intervention in the same group. However, in this case, it seems like two different groups (those on EFV-based ART and those on DTG-based ART) are being compared. These seem to be different sets of patients. This means that the same patient cannot be in both groups. This is therefore more akin to a cohort study. Please clarify the study design and consider revising it if necessary.

2.1.5. Use of health facility as randomization unit: The authors mention that “…. facilities ..…. were randomly selected to be included in the study”. The method or process of randomization is not clear. Was it simple randomization, block randomization, or stratified randomization? As it seems the randomization unit is the clinic, and since the method or process of randomization was not described, in the event that randomization was not effective, issues that could arise and need address could include: (a) selection bias if the population attending each clinic differs significantly in characteristics related to the outcome of interest, e.g age difference in between clinics, difference in health outcomes due to underlying population differences between the clinics; (b) confounding if there are any unmeasured confounding variables that differ between clinics and affect the outcome, e.g. quality of care, experience level of the staff of availability of certain resources; (c) cluster effects if patients within the same clinic are more similar to each other than to patients in other clinics; (d) problems with generalizability to all clinics or populations if the clinics included had specific characteristics that make them not representative of all clinics. There is therefore need to clearly describe the randomization method and process to understand if the above issues may have arisen.

2.1.6. Use of consecutive sampling: The authors mention that they used a consecutive sampling technique, but it’s not clear how this was implemented. Did they include all eligible participants until they reached their sample size? It is important to note that consecutive sample can introduce sampling bias depending on how it is done. It is therefore important for the authors to provide more details on how it was done and how they ensured that bias was not introduced. Furthermore, as the source of participants is an electronic database, the authors could consider randomly selecting participants from the data base instead of using consecutive sampling, which would ensure full randomization.

2.1.7. Threshold for covariate selection into the model: The authors selected covariates for entry into the multivariable model based on a p-value of less than 0.25 in the unadjusted analyses, which is not a standard approach as it may lead to overfitting. Though a more stringent or more conservative thresholds than the standard 0.05 can be used depending on the context, other factors may need to be taken into account as depending only on a p value can result in overfitting or underfitting of the model. Other statistical considerations that could be taken into account include the effect size, confidence intervals, and the clinical or practical significance of the covariates. It would therefore be helpful to provide a rationale for the choice of the threshold and what other consideration were taken into account.

2.1.8. Adherence categorization: The categorization of adherence into poor, fair, and good is not clearly defined. It would be helpful to provide specific criteria or thresholds for these categories.

2.1.9. Missing data: The results do not mention any missing data. Considering this study is using routine data, it would be very unusual for there to be no missing data. It’s therefore crucial to address this in the results and discuss how it might have affected the findings. However, if there actually were no missing data, the authors would need to explain how that was achieved.

2.1.10. Exclusion of variables statistically significant in table 4: Only the seven statistically significant variables are included in table 4. The authors could consider including those that were not statistically significant in table 4.

2.1.11. Confounding factors: The results do not discuss potential confounding factors. For example, were there any differences in the health status, lifestyle, or other characteristics between the two groups (DTG-based regimens arm and EFV-based regimens arm) that could have influenced the results?

2.1.12. Inconsistent comparisons: The authors mentioned that the study findings are in line with the China study (second discussion paragraph), and indicate the comparison regimen was protease-based, and put in brackets EFV-based. The China study was actually about low-level viremia and virological failure, and the comparison group was protease-based. The China study may therefore not be appropriate as a comparison group. Furthermore, viral load suppression was also compared with those of studies conducted in different settings using different viral load suppression thresholds, but they do not seem to take into account potential differences in study populations, methodologies or healthcare systems.

2.1.13. Unexplained contradictions: The authors state that older women were more likely to have non-suppressed viral load status but cite studies that found the opposite. The manuscript does not explain or discuss why there were these contradictions.

2.2. Minor issues

2.2.1. The following statements in the introduction may need review:

a. “Failure to achieve adequate VLS status ….. the major risk factor ….” does not flow well and may need to be rewritten.

It is usually high viral load that is the risk factor for MTCT of HIV.

b. In the second paragraph and first line of the introduction: It may be better to add to this phrase “putting women

on ART, preferably DTG-based regimens” to make the sentence complete.

2.2.2. Rationale for choosing study sites: The study setting is described, but the rationale for choosing these specific regions and zones is not clear. It would be helpful to provide a brief explanation for this choice.

2.2.3. Assumptions for sample size calculation: The sample size calculation is generally clear, but it would be helpful to provide more details about the assumptions used. For example, how was the ratio of unexposed to exposed determined to be one? Also, it would be helpful to provide a reference or explanation for the decision to add 20% for missing data.

2.2.4. Operational definitions: The operational definitions are clear, but it would be helpful to provide a bit more context. For example, why is a viral load count of less than or equal to 50 copies/ml considered suppression as it is not the standard definition as recommended by WHO (less than or equal 1,000 copies/ml). If the authors still prefer to use the non-standard cut-off for the study, it would be helpful to provide the specific rationale.

2.2.5. Referencing: The is issues with some of the referencing. For example, reference [10] is cited twice but it’s not clear if it refers to the same source.

2.2.6. Study setting: The authors mention that details on study setting are provided elsewhere. It would be beneficial to include at least a brief summary in this paper for completeness.

2.2.7. Effect of methodological limitations on results: The paper acknowledges methodological limitations. It however does not discuss how these limitations might have affected the results.

2.2.8. Lack of clarity on the study outcome: Though the title of the study clearly indicates it is on viral load non-suppression, some parts of the manuscript refer to viral load suppression so that the message of the study outcome is not consistent. It will help to harmonize this throughout the manuscript.

2.2.9. Numbers in the results section: Most of the numbers and percentages in the tables are repeated in the text. It would help readability and flow if only essential numbers and percentages are mentioned in the text with references being given to the tables. The results sections other than the one on factors associated with VL non-suppression do not refer to the variables found to meet the 0.25 p-value threshold for inclusion in the multivariate analysis.

2.2.10. Categorization of daily labourers as high risk: It is not clear why daily labourers are included in the high-risk group. The authors may need to explain this in the Ethiopian context.

2.2.11. Language and Grammar: There are some typos in the document and statement that need to be revised, e.g. “DGT-based regimen” and “Camerron” in the sample size section. There are also words that may need to be replace, for example: “adequate” in the abstract background, “employed” in the abstract methods, “supplementation” in the abstract conclusion; “besides” in the results section on factors associated with VL non-suppression; and the word “direction” in the third paragraph of the introduction section could be replaced with “Global targets”. There is also need for consistency in the use of some abbreviations, e.g. both “VLS” and “VL status” which could be harmonized throughout the manuscript. Overall, many of the sections do not flow well and need language and technical editing to flow better.

2.2.12. Strengthening the conclusion: The conclusion could be strengthened by summarizing the main findings of the paper and discussing their implications for clinical practice and future research. It would also be helpful to highlight key study limitations and their impact on the results.

6. PLOS authors have the option to publish the peer review history of their article (what does this mean?). If published, this will include your full peer review and any attached files.

Reviewer #1: No

Reviewer #2: **Yes: **Brian C Chirombo, MBChB, MPH

---

## [Author Response · Author response to Decision Letter 0]

17 May 2024

The authors are happy to get the opportunity to revise the manuscript to increase its quality. The authors have carefully considered each of the comments provided by Editor and two reviewers. The authors addressed all the comments, suggestions, clarify any ambiguities in the point-by-point response. The authors incorporated the comments and suggestions in the revised manuscript.

---

## [Editor Report · Decision Letter 1]

29 May 2024

Viral load non-suppression status among women exposed to Dolutegravir-based versus Efavirenz-based first-line antiretroviral therapy in Ethiopia: a  before-and-after study

PONE-D-24-09455R1

Dear Wolde Facha,

We’re pleased to inform you that your manuscript has been judged scientifically suitable for publication and will be formally accepted for publication once it meets all outstanding technical requirements.

Kind regards,

Moses Katbi

Academic Editor

PLOS ONE
---

## [Editor Report · Acceptance letter]

30 May 2024

PONE-D-24-09455R1 

PLOS ONE

Dear Dr. Facha, 

I'm pleased to inform you that your manuscript has been deemed suitable for publication in PLOS ONE. Congratulations! Your manuscript is now being handed over to our production team.

Kind regards, 

on behalf of

Dr. Moses Katbi 

Academic Editor

PLOS ONE